# *Vibrio cholerae* motility is associated with inter-animal transmission

Ian W. Campbell [1,3], Ruchika Dehinwal [1,3], Alexander A. Morano[1], Katherine G. Dailey[1], Franz G. Zingl [1] & Matthew K. Waldor [1,2] ✉

Outbreaks of cholera are caused by the highly transmissive pathogen *Vibrio cholerae*. Infant mouse studies have elucidated many aspects of *V. cholerae* pathogenesis; however, the components of pathogenesis that feed-forward to promote transmission have remained enigmatic because animal models routinely bypass the mechanisms of inter-animal transmission by directly inoculating cultured bacteria into the stomach. Here, a transposon screen reveals that inactivation of the *V. cholerae* motility-linked gene *motV* increases infant mouse intestinal colonization. Compared to wild-type *V. cholerae*, a Δ*motV* mutant, which exhibits heightened motility in the form of constitutive straight swimming, localizes to the crypts earlier in infection and over a larger area of the small intestine. Aberrant localization of the mutant is associated with an increased number of *V. cholerae* initiating infection, and elevated pathogen burden, diarrhea, and lethality. Moreover, the deletion of *motV* causes *V. cholerae* to transmit from infected suckling mice to naïve littermates more efficiently. Even in the absence of cholera toxin, the Δ*motV* mutant continues to transmit between animals, although less than in the presence of toxin, indicating that phenotypes other than cholera toxin-driven diarrhea contribute to transmission. Collectively, this work provides experimental evidence linking intra-animal bottlenecks, colonization, and disease to inter-animal transmission.

Cholera is a severe diarrheal disease caused by *Vibrio cholerae*, a highly motile Gram-negative rod[1]. Individuals become exposed to the pathogen by ingesting contaminated water or food, or through close person-to-person spread[2–4]. After ingestion, the pathogen colonizes and rapidly proliferates in the small intestine (SI). During its growth in the SI, *V. cholerae* secretes cholera toxin, an $AB_5$-type protein toxin whose activities largely account for the secretory diarrhea characteristic of cholera[5].

Orogastric inoculation of 3–5-day-old infant mice with toxigenic *V. cholerae* has been used to model cholera for over five decades[6–8]. The disease observed in infant mice mirrors many aspects of human infection, including toxin-dependent diarrheal disease[9]. Studies in this model have yielded many critical insights into the genes and processes that contribute to *V. cholerae* pathogenicity[10,11]. For example, toxin co-regulated pilus (TCP), the pathogen's signature colonization factor, was initially identified in infant mice[12] and was subsequently demonstrated to be critical for human infection[5].

Infant mouse studies also revealed that *V. cholerae* motility and chemotaxis play important and often opposite roles in intestinal colonization. The pathogen has a single polar sheathed flagellum whose counterclockwise rotation causes straight swimming. Chemotactic input causes random reorientation by briefly reversing the spin of the flagellum to clockwise rotation[13]. Nonmotile mutants lacking a flagellum or with an inactive flagellum (e.g., Δ*motAB*) exhibit marked defects in colonizing infant mice[14,15]. Conversely, chemotaxis mutants with an active flagellum often exhibit heightened colonization[15]. For

[1]Division of Infectious Diseases at Brigham & Women's Hospital, and Department of Microbiology at Harvard Medical School, Boston, MA, USA. [2]Howard Hughes Medical Institute, Boston, MA, USA. [3]These authors contributed equally: Ian W. Campbell, Ruchika Dehinwal. ✉e-mail: mwaldor@bwh.harvard.edu

example, *V. cholerae* chemotaxis mutants that cannot change the direction of flagellar rotation from counterclockwise to clockwise (e.g., Δ*cheY-3*) exhibit longer stretches of straight swimming and have increased intestinal colonization in infant mice, particularly in the proximal SI[16]. These *V. cholerae* motility and chemotaxis phenotypes have not been tested in infant rabbits, another model of *V. cholerae* pathogenesis[17], or in humans. However, it has been observed that *V. cholerae* shed in human feces are motile and transcriptionally repress several chemotaxis genes[18].

Cholera epidemics are often characterized as 'explosive' because of the rapid spread of the disease to exposed individuals[1]. Animal colonization models have elucidated many aspects of pathogenesis that could contribute to the characteristic transmissibility of *V. cholerae*, including pathogen colonization factors that facilitate replication within the intestine, a host-associated increase in pathogen infectivity (aka hyperinfectivity), and the role of virulence genes in diarrhea[10]. However, the mechanisms resulting in pathogen transmission are usually not captured in animal models because infections are initiated by directly inoculating cultured bacteria into the stomach, limiting our knowledge of the factors that govern *V. cholerae* transmissibility.

Here, we determine which *V. cholerae* genes modify colonization of infant mice using a transposon loss-of-function screen. We find that inactivation of *motV*, a mutation that increases *V. cholerae* motility in liquid[19], increases pathogen burden in the SI. Compared to wild-type *V. cholerae*, a Δ*motV* mutant evades the host bottleneck, resulting in more cells initiating infection by localizing to a broader range of the intestinal crypts, and causing more diarrhea. We leverage the phenotypes of the *motV* mutant to investigate how events within the intestine feed-forward to govern transmission by adapting the infant mouse model to directly quantify transmission between pups. We find that deletion of *motV* causes *V. cholerae* to transmit more efficiently, even in the absence of cholera toxin-dependent diarrhea. Thus, *V. cholerae* motility modifies its transmissibility.

## Results

### Genome-scale screen to identify *V. cholerae* genes modifying infant mouse colonization

Transposon insertion site sequencing (Tn-seq) is a powerful approach for genome-scale identification of bacterial genes that modify growth in diverse conditions[20–22]. In animal-based studies, Tn-seq screens are generally carried out to identify mutants that fail to colonize, suggesting that the transposon-disrupted gene promotes bacterial growth in the host[23–26]. However, Tn-seq has not yet been used to define the genes required for *V. cholerae* colonization of infant mice.

Colonization bottlenecks can render Tn-seq studies uninterpretable because the bottleneck causes a stochastic loss of mutants from the population, which can obscure the identity of mutants lost due to selective pressure in the host[27]. However, recent studies measuring the bottleneck restricting the *V. cholerae* population during colonization of infant mice suggest that Tn-seq is possible in this model; using genomically-barcoded *V. cholerae*, Gillman et al. (2021) demonstrated that when given a high dose (10^8 colony forming units, CFU), ~10^5 unique bacterial cells survive the bottleneck and expand in the SI of infant CD1 mice[28]. High-density Tn-seq libraries created with a mariner transposon generally contain ~10^5 unique mutants, approximately the same number of cells that survive the bottleneck. We used this insight to carry out a Tn-seq screen for *V. cholerae* genes that modify growth in the infant mouse intestine.

We orally inoculated ~5 × 10^7 CFU of a dense transposon library containing ~1.1 × 10^5 unique mutants created in a 2010 *V. cholerae* clinical isolate from Haiti[29] into postnatal day 4 Crl:CD1(ICR) mice (henceforth P4 CD1). 18-hours after inoculation, we recovered between 1.7 to 2.5 × 10^4 of 1.1 × 10^5 unique mutants from the SI of 3 pups (Fig S1A), indicating that the bottleneck caused a substantial loss of

mutant diversity. To increase the recoverable mutants, we pooled the SIs of 10 P4 mice (one litter) and recovered 3.3 × 10^4 unique mutants, covering ~35% of the average gene's potential mutagenesis sites (Fig S1A, B). This degree of mutant loss can be accommodated by our analytical pipeline, which uses simulation-based normalization to model and compensate for the random loss of diversity caused by population bottlenecks[27,30].

Using this approach, we identified the genes that promote *V. cholerae* survival in the infant mouse SI (Supplementary Data 1). Transposon insertions in genes that facilitate growth in the murine intestine but not when cultured in LB media are found on the left side of the volcano plot in Fig. 1A. The hits on the left side of the plot validate this application of Tn-seq. Transposon insertions in many genes previously demonstrated to be critical for intestinal colonization in this model, such as genes required for the biogenesis of the type IV pilus TCP (shown in orange in Fig. 1A, B), had markedly reduced abundance in vivo. Within the *tcp* locus, the reduction in the abundance of insertions in genes such as *tcpA*[31], the major subunit of the pilus, and *tcpF*, a secreted and essential colonization factor[32], was greater than 1000-fold; in contrast transposon insertions were not depleted in *tcpI* during colonization, a gene that is not required for colonization and that has been linked to reduced TcpA expression[33], illustrating the specificity of the Tn-seq screen.

Flagellar-based motility is generally thought to promote *V. cholerae* intestinal colonization in infant mice[13], and insertions in many of the genes linked to the biogenesis and spinning of the pathogen's flagellum had reduced abundance in animal samples relative to growth in LB (Fig. 1A, B). In contrast, there was no reduction in insertions in the flagellin genes *flaABCDE*, which are the building blocks of the flagellar filament. The lack of a phenotype associated with disruption of *flaABCDE* may be due to functional redundancy between the flagellins. However, unlike *flaBCDE*, *flaA* was previously determined to be necessary and sufficient for motility in another *V. cholerae* strain[34], and it is unclear why *flaA* is not required for colonization here.

Intriguingly, several genes were also identified on the right side of the volcano plot (Fig. 1A). Insertions in these genes were more abundant in vivo than in vitro, suggesting that these genes antagonize *V. cholerae* growth in the intestine. Within the initial LB library, no individual mutant was represented by more than 0.02% of sequencing reads. Mutants that became more abundant in the animals were visualizable as peaks in transposon abundance, with individual mutants comprising >0.4% of reads (Figs. 1C, S1C). Four of these genes, *cheA−2*, *cheR-2*, *cheV-3*, and *cheY-3* (Fig. 1A), are linked to chemotaxis, which was previously demonstrated to limit colonization of infant mice[13]. *motV* insertions were among the most enriched in vivo (Fig. 1A, C). *motV* is encoded by a small subset of flagellated bacteria[19] and the molecular function of MotV is unknown. MotV was previously demonstrated to control motility and associate with HubP[19], the *V. cholerae* cell pole organizer[35]. Without *motV*, *V. cholerae* cannot reverse directions (tumble). Cells lacking *motV* have increased motility in liquid, constitutively swimming straight at elevated speeds. Conversely, cells lacking *motV* cannot swim in soft agar, which requires reversing direction[19].

MotV's role in *V. cholerae* infection has not been studied; no information indicates whether this gene has a role in human infection, and defined *motV* mutants have not been tested in the infant mouse or infant rabbit models. Furthermore, Tn-seq studies in infant rabbits differ on the role of *motV* during infection[27,36–38], indicating a need for further investigation.

### Deletion of *motV* elevates *V. cholerae* intestinal colonization

To corroborate that *motV* antagonizes *V. cholerae* colonization in infant mice, we created a *motV* deletion mutant (Δ*motV*). As expected, the Δ*motV* strain had severely impaired motility in soft agar, like that observed in a Δ*cheY-3* mutant (Fig S2A, B), and the provision of a

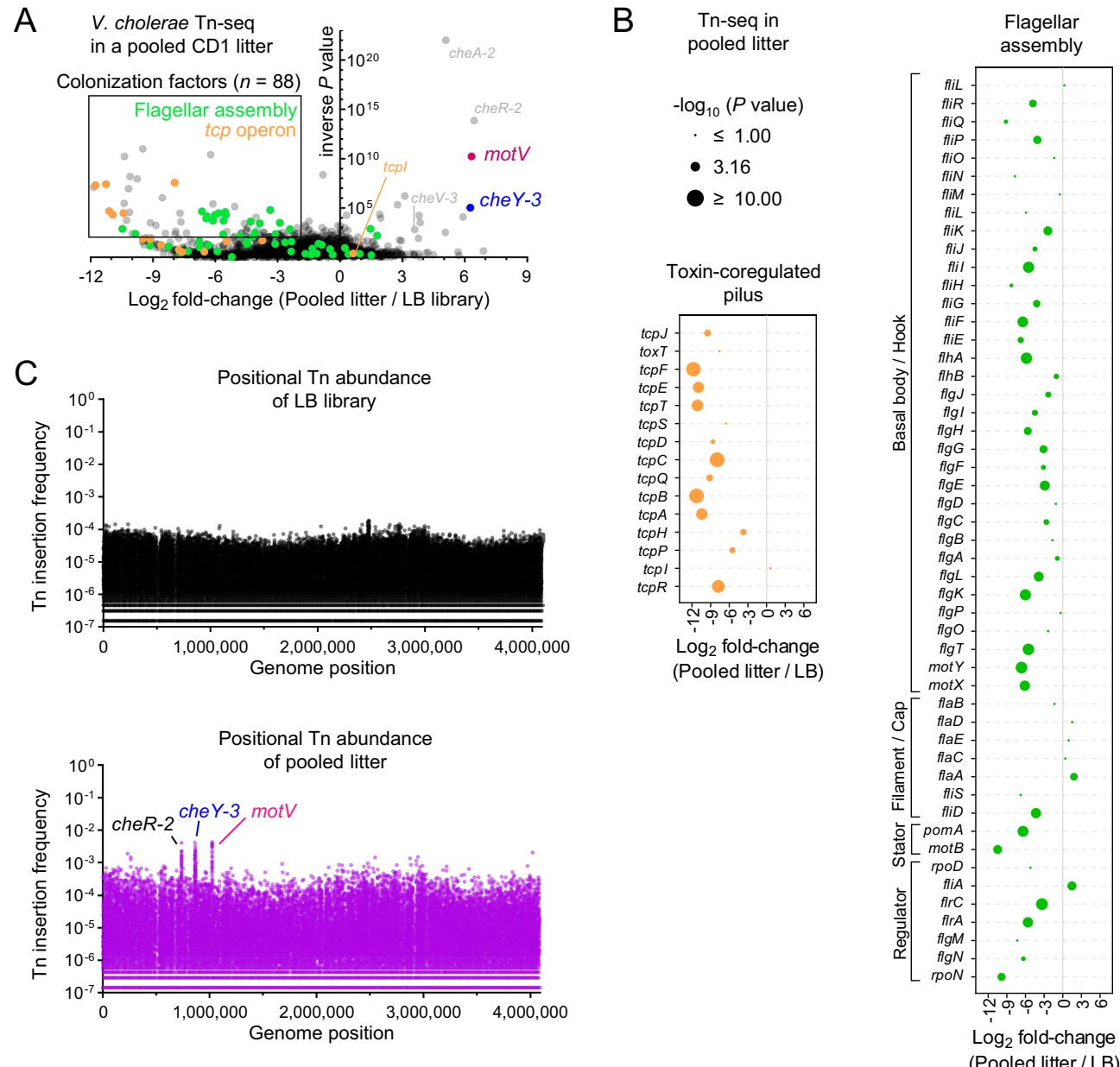

**Fig. 1 | Tn-seq screen reveals that disruption of *motV* increases *V. cholerae* infant mouse colonization.** 10 postnatal day 4 Crl:CD1(ICR) mice (henceforth P4 CD1) were intragastrically inoculated with ~5 × 10⁷ CFU of a mariner transposon library. 18 h later, the SIs were homogenized and pooled, and after overnight growth on LB plates, transposon insertion sites were determined by sequencing (pooled litter). For comparison, the same library was passaged overnight on LB plates (LB library). *n* = 1 pooled litter of 10 pups and 1 LB plate. **A** Volcano plot comparing the fold-change in gene insertion frequency between the pooled litter and LB library with *P* value derived from a two-sided Mann-Whitney test. Colonization factors are genes with *P* value < 0.01 and log₂ fold-change <−2. Data for the *tcp* operon (toxin-coregulated pilus) and flagellar assembly genes (categorized from KEGG database vch02040) are expanded in (**B**). **C** Transposon insertion frequency across the genome highlighting increased insertion frequency in *cheR-2, cheY-3*, and *motV* from the pooled intestinal samples. Tn-seq data are available in Supplementary Data 1. Source data are provided as a Source Data file.

plasmid-borne copy of *motV* (p*motV*) restored soft agar motility (Fig S2C). While immobile in soft agar, *motV* and *cheY-3* mutants have increased motility in liquid[19]. We employed a mucus penetration assay[39–41] to investigate how the lack of *motV* may impact motility within the intestine. Both Δ*motV* and Δ*cheY-3* mutants had increased motility, migrating further into a 1% mucus column than wild-type (WT; Fig S2D).

In infant mice, the number of Δ*motV* CFU recovered from the SI exceeded the CFU recovered from animals infected with WT *V. cholerae* and was similar to the CFU found with a Δ*cheY-3* mutant (Fig. 2A), which is known to increase colonization[15]. Further analyses found that the heightened colonization of the Δ*motV* strain

was particularly prominent in the proximal SI, where Δ*motV* CFU exceeded those of the WT by ~13-fold (Fig. 2B). Ordinarily, ~10x more WT *V. cholerae* CFU are recovered from the distal SI than the proximal SI. However, similar CFU of the Δ*motV* mutant strain were recovered from the two parts of the intestine. The elevated capacity of the Δ*motV* mutant to colonize the proximal SI was reduced by the provision of p*motV* (Fig S2E), suggesting that this Δ*motV* phenotype is specifically caused by the deletion of *motV*. Moreover, the heightened colonization capacity of the Δ*motV* strain was confirmed in competition assays, where *lacZ*⁺ Δ*motV* outcompeted a *lacZ* WT strain 13-fold in the proximal SI and 4-fold in the distal SI (Fig. 2C).

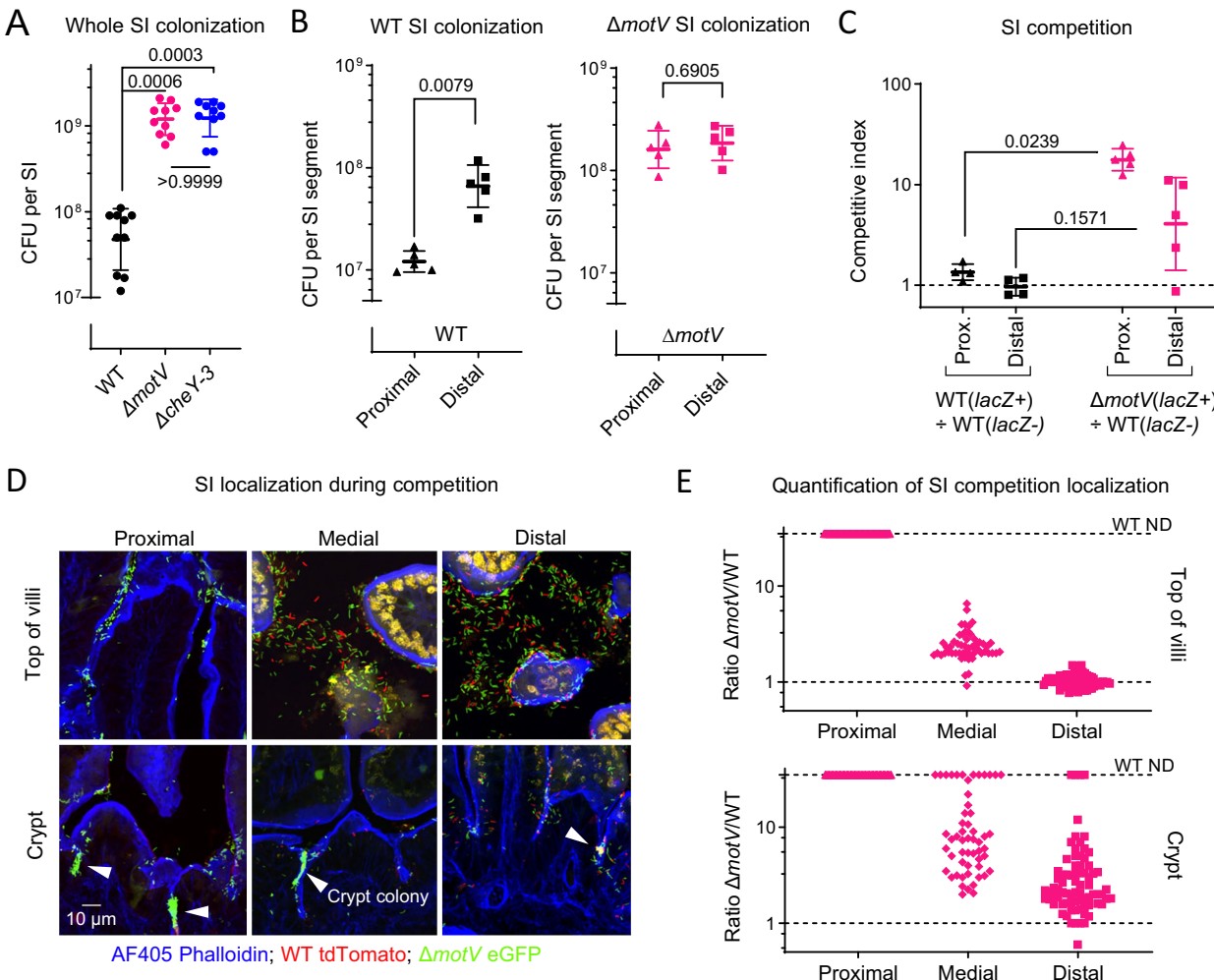

**Fig. 2 | Deletion of *motV* increased *V. cholerae* SI colonization, especially in the proximal SI and within the crypts.** CD1 pups were intragastrically inoculated with ~3 × 10⁶ CFU of the indicated *V. cholerae* strains, and 18-hours later the whole intestine (**A**) or the indicated SI segments (**B**) were plated on selective media to determine *V. cholerae* colony-forming units (CFU). **C** CD1 pups were infected with ~2 × 10⁶ CFU of a ~1:1 ratio of *lacZ⁺* to *lacZ⁻ V. cholerae*, and the competitive index was calculated as the ratio of *lacZ⁺* to *lacZ⁻* CFU in the indicated SI segment 18-hours after inoculation divided by the ratio of *lacZ⁺* to *lacZ⁻* CFU in the inoculum. **A** *n* = 30 pups (3 litters) randomized between strains. **B** *n* = 10 pups (1 litter) per strain. **C** *n* = 9 pups (1 litter) split 4 WT/WT and 5 Δ*motV*/WT. **A**, **C** Kruskal-Wallis test with Dunn's multiple comparison correction.

**B** Two-sided Mann-Whitney test. **A**–**C** Geometric mean and standard deviation. **D** Images of 10 μm cryosections of the indicated SI sections 18-hours after inoculation with ~2 ×10⁶ CFU of a ~1:1 ratio of WT *lacZ::tdTomato V. cholerae* (red) and Δ*motV lacZ::eGFP V. cholerae* (green). Sections were stained with AF405-conjugated phalloidin (blue) and imaged on a spinning disk confocal microscope. Arrowheads show *V. cholerae* microcolonies. **E** Quantification of the ratio of Δ*motV*:WT *V. cholerae* cells at the top(s) of the villi and in the crypts of the SI. When WT was not detected (ND), the ratio was given a value of 35 as an upper limit. *V. cholerae* were differentiated from background fluorescence by comma/rod morphology. 50–60 fields per site, imaged from the SIs of 4 mice. Source data are provided as a Source Data file.

A 1:1 ratio of eGFP-tagged Δ*motV* and tdTomato-tagged WT *V. cholerae* were co-inoculated into infant mice to determine if *motV* deletion changed the pathogen's localization within the SI. Notably, previous work found that fluorophore expression does not measurably alter *V. cholerae* fitness[14]. There was a much greater abundance of the Δ*motV* strain compared to the WT strain, particularly in the proximal SI, where it was difficult to detect the WT strain (Fig. 2D, E). Besides corroborating the competitive advantage of the Δ*motV* mutant along the proximal to distal axis of the SI, the fluorescence microscopy also revealed a differential localization of the two strains along the crypt-villus axis. In all segments of the SI, the Δ*motV* mutant had an enhanced capacity to access the crypts and form microcolonies (Fig. 2D, E; Fig S3), consistent with the increased motility of the Δ*motV* mutant in mucus (Fig S2D). Thus, the absence of *motV* enhances colonization at least in part by facilitating the pathogen's entry into the SI crypts, especially in the proximal SI.

## The Δ*motV* mutant localizes early to the SI crypts

To gain insight into how deletion of m*otV* augments *V. cholerae* intestinal colonization, we compared the WT and mutant populations throughout the course of infection. There was a pronounced collapse in the size of the WT population over the first 3 h of infection, especially in the proximal SI (Fig. 3A), as previously described ref. 42. After the first 3 h, the WT population expanded in the proximal, medial, and distal SI. Notably, the size of the Δ*motV* mutant population was much less constricted over the first 3 h, particularly in the proximal SI. Subsequently, the Δ*motV* population remained more numerous than the WT population in the proximal SI from 3 to 24 hours, while the populations approached parity in the medial and distal SI at later timepoints.

We hypothesized that the constitutive straight swimming motility phenotype of the Δ*motV* mutant might contribute to the pathogen's capacity to persist in the proximal SI. To test this hypothesis, we used

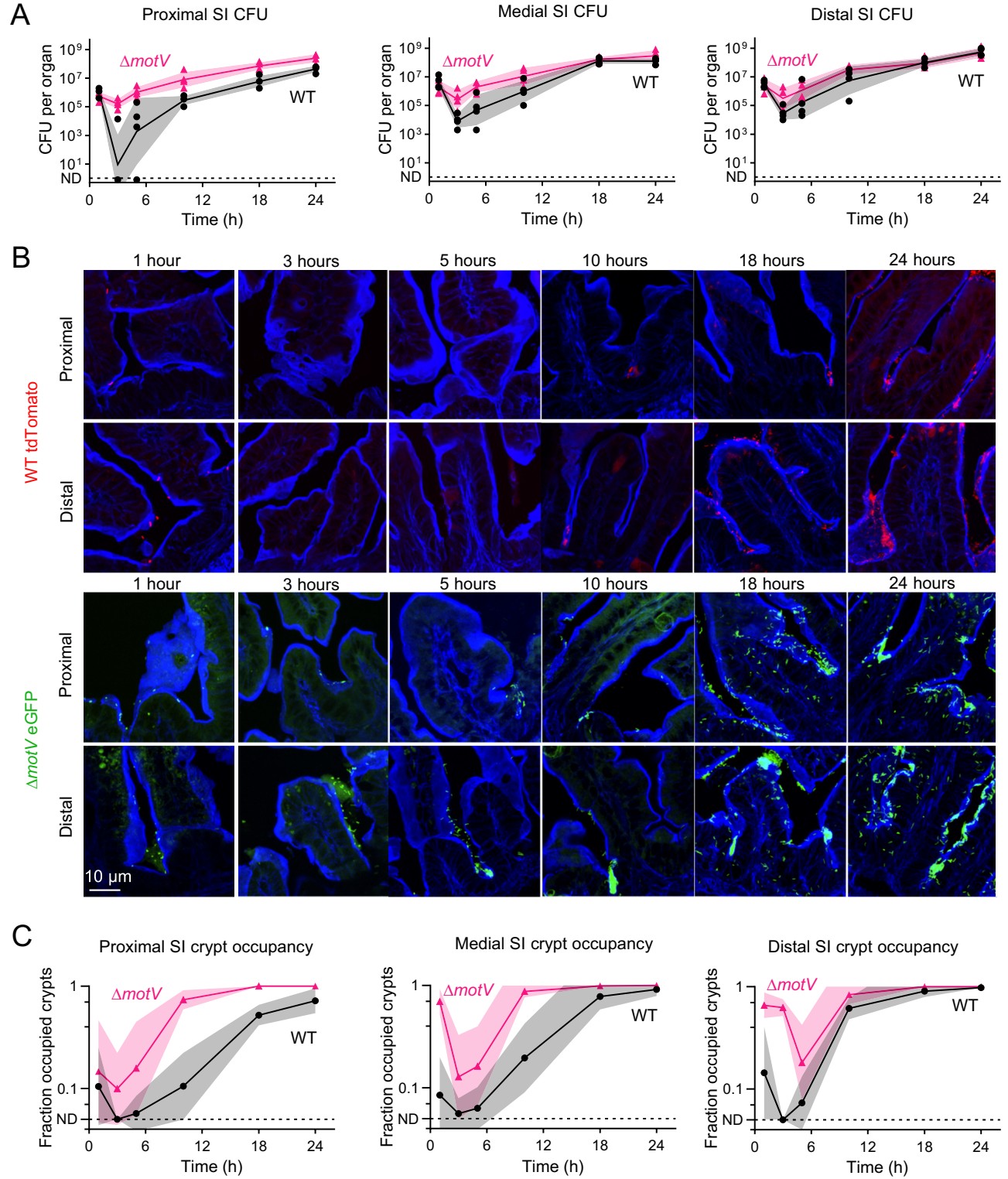

**Fig. 3 | The Δ*motV* mutant localizes in the SI crypts earlier during infection than WT *V. cholerae*.** CD1 pups were intragastrically inoculated with ~2 × 10⁶ CFU of either WT *lacZ::tdTomato V. cholerae* (red) or Δ*motV lacZ::eGFP V. cholerae* (green). CFU (**A**) and cryosections were collected over 24-hours from the indicated SI sections. Sections were stained with AF405-conjugated phalloidin (blue) and imaged on a spinning disk confocal microscope (**B**). **C** Quantification of the number of crypts per field occupied by at least one *V. cholerae* in the proximal, medial, and distal SI. ND, *V. cholerae* not detected in any field. *n* = 48 pups (5 litters) randomized with 4 animals per timepoint per strain. **C** *n* = 20 fields per timepoint per strain, **A**, **C** Geometric mean and standard deviation. Source data are provided as a Source Data file.

fluorophore-labeled *V. cholerae* to measure the pathogen's localization throughout the course of infection (Fig. 3B). Within the first hour, when the total burden was similar for both strains (Fig. 3A), it was already apparent that the Δ*motV* mutant was more numerous in the intervillous crypts. We quantified the fraction of crypts per field that were occupied by at least one *V. cholerae* at each timepoint (Fig. 3C). Throughout the infection, a greater fraction of crypts in the proximal small intestine were occupied by Δ*motV* than WT. Greater crypt occupancy by the Δ*motV* mutant was also observed in the medial and distal SI for most of the infection, with wild-type reaching parity at 24-

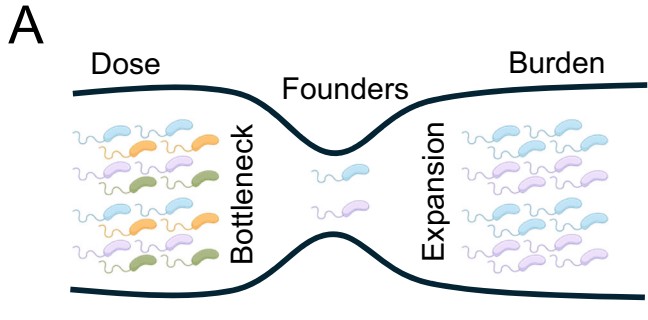

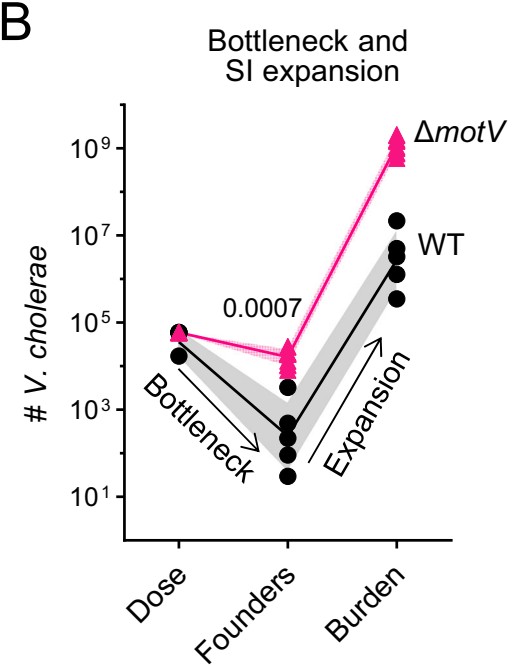

**Fig. 4 | *V. cholerae* lacking *motV* evaded the colonization bottleneck. A** Diagram of bacterial population dynamics during infection. The dose of the inoculum and the burden of the pathogen in tissue is enumerated by plating for CFU on selective media. The founding population is the number of unique cells from which the observed population originated and is measured by the loss of barcode diversity (represented as colors) between the inoculum and the observed population. Founders are calculated by STAMPR and expressed as Nr. The bottleneck is the loss of cells between the dose and founders, and net expansion is the gain between founders and burden. **B** Dose, founders, and burden of WT and *ΔmotV V. cholerae* in the SI of CD1 pups 18 h post-inoculation. WT, *n* = 5 pups (1 litter); *ΔmotV*, *n* = 10 pups (1 litter). Mann-Whitney test. Geometric mean and standard deviation. Source data are provided as a Source Data file. Parts of this figure were created in BioRender. Campbell, I. (2025) https://BioRender.com/egsdhsd.

hours. The increased penetration of the *ΔmotV* mutant into the crypts may result from its constitutive straight swimming and suggests that the motility behavior of the wild-type strain is insufficient to efficiently reach the crypt niche, particularly early in infection.

## The *ΔmotV* mutant circumvents the host bottleneck

We hypothesized that the failure of WT *V. cholerae* to reach the crypts and persist in the proximal SI during the first hours following inoculation may contribute to the initial collapse of the population and account for at least a portion of the WT colonization bottleneck. To directly characterize how deletion of *motV* impacts the *V. cholerae* colonization bottleneck, we used 'STAMPR', a lineage tracing method

that tracks bacterial population dynamics during infection using otherwise genetically identical cells identified by unique DNA barcodes[43]. Comparing the frequency and diversity of barcodes in the inoculum to samples from host tissues enables measurement of the number of unique bacterial cells (founding population) from which the observed population originated (Fig. 4A). The change in the number of cells from the inoculum to the founding population quantifies the infection bottleneck - the set of factors (e.g., physical barriers, immune processes, and the microbiota) that limit the capacity of a bacterial population to establish infection, with smaller founding populations reflecting tighter (more restrictive) infection bottlenecks.

The increased burden of the *ΔmotV* mutant in the SI compared to WT could be due to the strain undergoing a less restrictive bottleneck, leading to a larger founding population and/or more net replication. To distinguish between these possibilities and quantify the *ΔmotV* bottleneck, we used STAMPR to compare the population dynamics of barcoded WT and *ΔmotV* bacteria in P5 mice. When administered similar-sized inoculums, the founding population of the *ΔmotV* strain in the SI was ~100-fold greater than that measured for the WT strain (Fig. 4B), indicating that the mutant experienced a much less restrictive bottleneck. Surprisingly, the *ΔmotV* strain only experienced a ~2-fold bottleneck, with half of the *ΔmotV* cells in the inoculum surviving to become the founding population (Fig. 4B). Thus, the absence of *motV* allows *V. cholerae* to evade the factors that underlie the infection bottleneck. We propose that the ability of the *ΔmotV* mutant to evade host bottlenecks is caused by its constitutive straight swimming phenotype, which allows cells to persist in the proximal SI and invade deeper into the intestinal crypts (Fig. 3), forming replicative microcolonies (Fig. 2D).

## Deletion of *motV* increases *V. cholerae* virulence

We observed that the infant mouse bedding was more stained with diarrheal discharge in animals infected with the *ΔmotV* mutant than those infected with the WT strain (Fig S4), suggesting that the *ΔmotV* mutant induced more diarrhea in addition to increasing colonization. Indeed, the weight of the bedding, a measure of the diarrheal discharge, was greater in the *ΔmotV* group than WT (Fig. 5A). Furthermore, the mice infected with the mutant strain had more weight loss and a greater ratio of SI/body weight (a measure of fluid accumulation) than mice infected with the WT strain (Fig. 5A). Although the *ΔcheY-3* mutant colonized at least as well as the *ΔmotV* mutant (Fig. 2A), it did not lead to increased diarrheal discharge or weight loss (Fig. 5A). Thus, the increased colonization of the *ΔmotV* strain likely does not entirely account for its elevated diarrheaogenicity. The elevated virulence of the *ΔmotV* strain was also apparent in survival studies[9,44], where pups were returned to dams, and morbidity was monitored until a pre-determined 30-hour endpoint. The *ΔmotV* mutant resulted in mortality in infected animals more rapidly than either the WT or *ΔcheY-3* strains (Fig. 5B).

The principal cause of *V. cholerae*-associated diarrhea is cholera toxin[5]. To test whether deletion of *motV* increases diarrheaogenicity by causing cells to produce more toxin, we measured toxin production in laboratory conditions that induce cholera toxin synthesis (AKI)[45,46]. The WT, *ΔmotV*, and *ΔcheY-3* strains produced similar quantities of cholera toxin in these conditions (Fig S5). Since both the *ΔcheY-3* and *ΔmotV* mutants exhibit defective motility characterized by an inability to tumble and swim in soft agar[19] (Fig S2), increased burden during infection (Fig. 2A), and similar production of cholera toxin in culture, the cause of the increased virulence of the *ΔmotV* mutant remains unclear.

## The *ΔmotV* mutant is hyper-transmissible

Many of the phenotypes of the *ΔmotV* mutant are associated with transmission, including bottleneck evasion, replication to high burdens in the intestine, and shedding into the environment in diarrhea. To experimentally test whether these characteristics feed-forward to

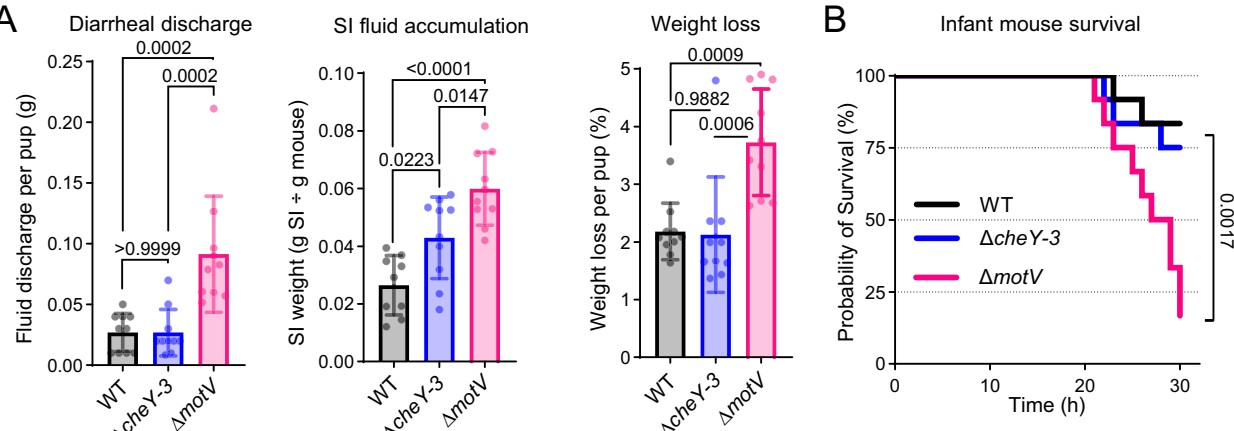

**Fig. 5 | Deletion of *motV* increased *V. cholerae* diarrheaogenicity. A** CD1 pups were intragastrically inoculated with the indicated *V. cholerae* strains and individually housed for 18-hours. The bedding was weighed prior to and after infection to determine fluid discharge. Fluid accumulation in the SI 18-hours post inoculation was measured by comparing the weight of the animal to the weight of the SI. Weight loss was determined by comparing animal weight before and after infection. *n* = 30 pups (3 litters) randomized between strains. Mean and standard deviation. One-way ANOVA with Tukey's multiple comparison correction. **B** CD1 pups were intragastrically inoculated with the indicated *V. cholerae* strains, returned to dams for care, and monitored until the predetermined 30-hour endpoint. Survival kinetics with two-sided Mantel-Cox test. *n* = 36 pups (4 litters) randomized between strains. Source data are provided as a Source Data file.

increase spread between animals, we devised a method to quantify *V. cholerae* transmissibility (Fig. 6A). In this system, post-natal day 4 mice from several litters were randomized into new litters to minimize litter bias. The next day, ~1/3 of the mice in each new litter were infected with the indicated *V. cholerae* strain; these infected 'seed' mice were then mixed with approximately twice as many uninfected 'contact' mice (the remaining 2/3 of the new litter) and returned to foster dams in individual cages for 20 h (Fig. 6A). At that point, the number of CFU in the SI of seed and contact groups was quantified to determine the number of infected contacts and the robustness of their colonization.

Across three independent trials, seed animals infected with the Δ*motV* mutant were more transmissive than WT (Fisher's exact test *P* value 0.0002); 13/18 (72%) of contacts of Δ*motV* seeds were infected, whereas 2/19 (11%) of contacts of WT seeds were infected (Fig. 6A). Additionally, contact animals that were infected by Δ*motV* seed animals tended to have greater intestinal colonization than contacts of the WT seed animals (Fig. 6C). Thus, the Δ*motV* mutant is 'hyper-transmissible', indicating that MotV impedes *V. cholerae* transmission as well as colonization. To our knowledge, the Δ*motV* strain represents the first description of a hyper-transmissible *V. cholerae* mutant.

We leveraged the Δ*cheY-3* mutant to test whether the hyper-transmissibility of the Δ*motV* mutant is solely attributable to its increased intestinal colonization. Like *motV* mutants, strains lacking *cheY-3* exhibit increased colonization (Fig. 2A and refs. [14–16]). However, the Δ*cheY-3* mutant was not hyper-transmissible in this assay, transmitting to fewer contacts (4/19; 21%) than Δ*motV* (Fig. 6A). These results suggest that the increased colonization of Δ*motV* does not fully account for its enhanced transmission.

Since the hyper-transmissibility of the Δ*motV* mutant is not solely attributable to increased colonization, we hypothesized that increased transmission could be caused by increased shedding of the pathogen in the more abundant diarrhea of mice infected by the mutant, a phenotype not shared with Δ*cheY-3*. To test this hypothesis, we used a Δ*ctxAB* mutant that does not produce cholera toxin, the principal cause of *V. cholerae*-associated diarrhea[5]. As expected, mice infected with strains lacking *ctxAB* or lacking both *ctxAB* and *motV* had less diarrheal discharge and less fluid accumulate in their SI than mice infected with the Δ*motV* mutant (Fig S6). There was no transmission from seed animals infected with a strain lacking *ctxAB* (0/18; 0%; Fig. 6A), suggesting that cholera toxin promotes the transmission of WT *V. cholerae* in this system. Furthermore, seed animals infected with

the Δ*motV* Δ*ctxAB* double mutant were less transmissive than seed mice infected with *V. cholerae* lacking *motV* alone (Fig. 6A, C), suggesting that cholera toxin-driven diarrhea contributes to transmission of the Δ*motV* mutant.

Nevertheless, cholera toxin was not required for the Δ*motV* mutant to transmit, as the Δ*motV* Δ*ctxAB* double mutant transmitted to 6/18 (33%) of contacts (Fig. 6A). Notably, since seed animals infected with the Δ*motV* Δ*ctxAB* double mutant were more transmissive than seeds infected with a Δ*ctxAB* single mutant, a portion of the Δ*motV*'s hyper-transmissibility is independent of cholera toxin-induced diarrhea.

## Discussion

A hallmark of infectious agents is their capacity for transmission between hosts. However, there is a relative paucity of experimental models to measure pathogen transmissibility, limiting our knowledge of the genes and processes involved in transmission. Here, we found that suckling mice infected with *V. cholerae* could transmit the pathogen to uninfected littermates, providing an experimental model of transmission. We leveraged this model to characterize how the motility-linked gene *motV* modifies *V. cholerae* transmission. The Δ*motV* mutant localized to the intestinal crypts earlier and over a larger area of the small intestine than wild-type *V. cholerae*. This change in localization was associated with 3 phenotypes that likely drive the hyper-transmissibility of the Δ*motV* mutant: (1) evasion of the host bottleneck, (2) elevated SI colonization, and (3) increased diarrheaogenicity.

Our findings suggest a model to describe how the constitutive straight swimming phenotype of the Δ*motV* mutant may result in hyper-transmissibility (Fig. 7). While WT chemotaxis inefficiently localizes *V. cholerae* to the crypt niche during the first hours following inoculation (Fig. 3B-C), deletion of *motV* enables earlier and more efficient occupancy of the intestinal crypts, especially in the proximal SI. We propose that the aberrant pattern of localization of the Δ*motV* mutant underlies many of its other phenotypes. Thus, early crypt occupancy may enable the Δ*motV* mutant to escape peristaltic flow and linger longer in the proximal SI, potentially explaining why the Δ*motV* mutant experiences a less restrictive bottleneck. Furthermore, occupying more of the SI likely enables the Δ*motV* strain to reach a higher burden, and the higher burden combined with proximity to the intestinal epithelium likely contributes to greater intoxication of the

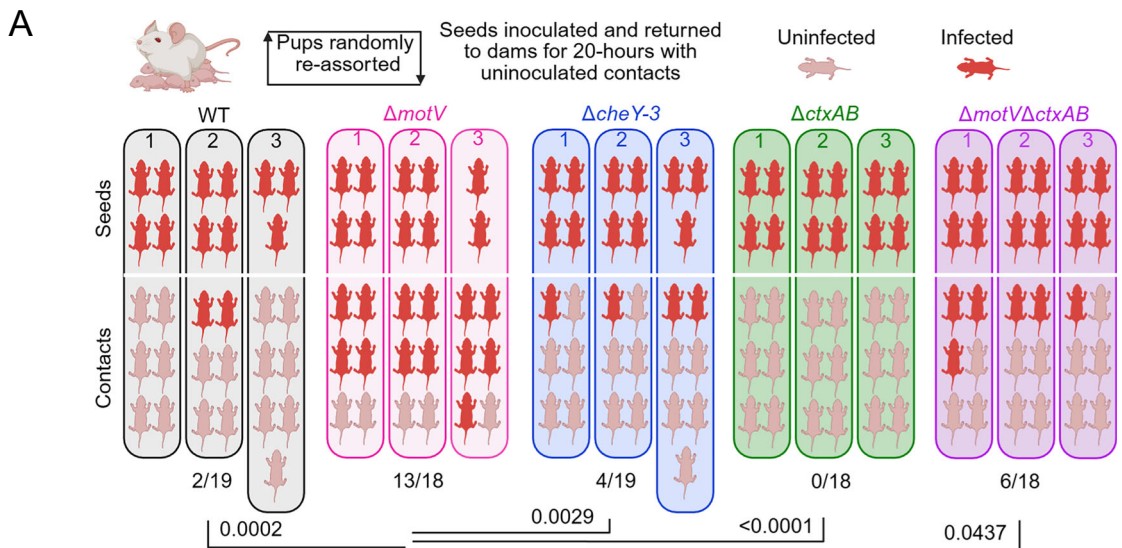

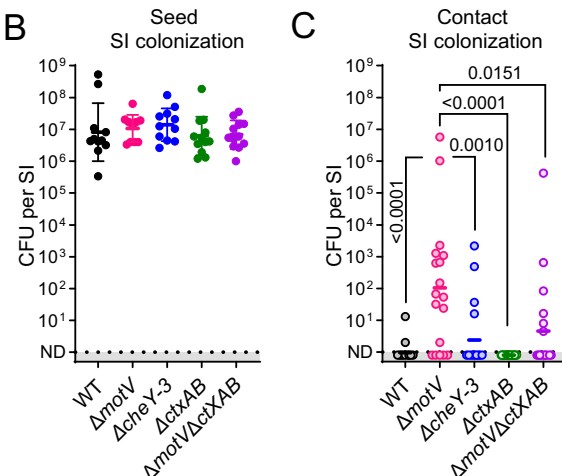

**Fig. 6 | Deletion of *motV* increased inter-animal transmission.** Transmission assays were performed by randomly reassorting CD1 pups, challenging ~1/3 of the litter with the indicated strain of *V. cholerae* (seeds), and returning them to foster-dams with naïve littermates (contacts). 20 h later, transmission was determined by enumerating CFU in the SI. Animals per group are shown in (**A**). Experimental groups: WT, Δ*motV*, and Δ*cheY-3* n = 88 pups (9 litters) randomized between strains; Δ*ctxAB* and Δ*motV* Δ*ctxAB* n = 60 pups (6 litters) randomized between strains. **A** Experimental results divided by litter. Uninfected contacts, 0 CFU detected; infected, ≥1 CFU identified. Two-sided Fisher's exact test. **B** Seed SI CFU determined 20 h post inoculation. Geometric mean and standard deviation. **C** Contact SI colonization determined after 20 h of cohousing with seed animals. Geometric mean. Kruskal-Wallis test with Dunn's multiple comparison correction. Source data are provided as a Source Data file. Parts of this figure were created in BioRender. Campbell, I. (2025) https://BioRender.com/7ac9055.

host and increased pathogen shedding. Moreover, shed Δ*motV V. cholerae* avoids the bottleneck and colonizes the next host at a lower dose. Collectively, aberrant motility leads to more cells from the inoculum initiating infection, elevated burden, and increased diarrhea, which together feed-forward to heighten transmission. Our findings illuminate the processes that are likely exploited by many enteric pathogens for their characteristic transmission: (1) evasion of host barriers, (2) replication to high burdens in infected tissues, and (3) dissemination in diarrhea, and demonstrate how intra-animal pathogenesis results in inter-animal transmission.

We observed that the Δ*motV* mutant had an unusual property – it largely escaped the host infection bottleneck (Fig. 4). In general, we interpret bottlenecks as host barriers to infection, and that host and/or microbiota changes can modulate the bottleneck. For example, the low pH of the stomach and the presence of an intact microbiota both contribute to the bottleneck restricting enteric colonization by the murine pathogen *Citrobacter rodentium*[47,48]. The *motV* mutant is a new

example of an emerging paradigm[49,50] that pathogen factors contribute to the bottleneck.

We propose that the circumvention of the bottleneck by the Δ*motV* mutant is a consequence of its aberrant swimming behavior and greater localization to the SI crypt niche. Paradoxically, by this logic, MotV activity in wild-type *V. cholerae* subjects the pathogen to more stringent population restriction. We observed that deletion of *motV* increased *V. cholerae* motility in mucus (Fig S2D). Previous work determined that mucus is a potent barrier to *V. cholerae* colonization of the infant mouse proximal SI[14], the SI segment with the largest increase in colonization caused by deletion of *motV*. These results may reveal that a major underlying contributor to the *V. cholerae* bottleneck in CD1 infant mice is a failure of wild-type cells to penetrate the mucus in the proximal SI, resulting in the removal of bacteria by intestinal peristalsis.

By escaping the colonization bottleneck, bacteria lacking *motV* remain at a higher burden throughout the small intestine for the entire

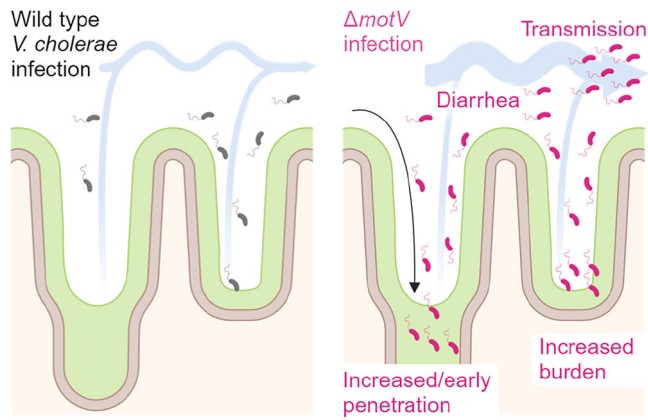

**Fig. 7 | Model connecting the heightened inter-animal transmission of ΔmotV V. cholerae to aberrant motility, bottleneck evasion, increased burden, and increased diarrheaogenicity.** We propose that the ΔmotV mutant's constitutive and rapid motility enables V. cholerae to penetrate deeper into the more proximal crypts of the SI early during infection and that this aberrant pattern of localization accounts for its hyper-transmissibility. The expansion of the V. cholerae intestinal niche permits more of the inoculum to survive the colonization bottleneck and replicate, increasing the burden of V. cholerae in the intestine and leading to greater diarrhea. Greater burden and diarrheal flux facilitate the excretion of the pathogen into the environment, promoting transmission. Created in BioRender. Campbell, I. (2025) https://BioRender.com/p24d207.

infection. The increased colonization of ΔmotV V. cholerae, particularly in the proximal SI, is similar to that observed with cheY-3 chemotactic mutants (Fig. 2A-C)[14–16]. Although motV is not known to be linked to the output of V. cholerae's complex chemoreceptors and chemosensory pathways, deletion of motV and cheY-3 cause similar motility phenotypes[19]. Cells lacking motV or cheY-3 are both only capable of counterclockwise flagellar rotation; they are unable to reverse the direction of flagellar rotation from counterclockwise to clockwise and consequently show increased smooth swimming in straight lines, decreased tumbling, and are unable to swim in soft agar[16,19]. Studies using fluorescently labeled chemotactic deficient V. cholerae suggested that chemosensory pathways are not required for V. cholerae to penetrate into the intestinal crypts or for the localization of the pathogen along the crypt-villus axis[14]. The similar swimming behavior and colonization phenotypes of the cheY-3 and motV mutants suggest that their constitutive straight swimming phenotype may cause their shared capacity for heightened colonization of the proximal SI.

We propose that the aberrant swimming behavior and greater localization of the ΔmotV mutant to the small intestine crypts increases disease in infected hosts. The swimming behavior of ΔmotV may increase diarrheaogenicity by bringing the pathogen into a virulence-stimulating location and/or increasing the efficacy of toxin delivery by locating the pathogen closer to the epithelium. Moreover, the increased pathogen burden in animals infected with the ΔmotV mutant likely also increases the quantity of toxin production, although elevated pathogen burden alone cannot fully explain the heightened diarrhea as cheY-3 mutants have similar increased colonization without increasing diarrhea. While the molecular mechanism linking motV to intoxication remains to be described, the difference in diarrheaogenicity of the cheY-3 and motV mutants provided a tool to probe the role of toxigenic diarrhea in V. cholerae transmission.

The ΔmotV mutant's increased colonization, virulence, evasiveness, and transmission could all be linked to its aberrant swimming behavior. Although motility is important in V. cholerae virulence, many pathogenic bacteria have lost their flagellum (e.g., Shigella spp., Mycobacterium tuberculosis, Klebsiella pneumoniae) or suppress expression of their flagellum during infection (e.g., Listeria

monocytogenes)[51–55]. However, among the pathogens that have retained the flagella, there is evidence that motility and chemotaxis play a role during infection[13]. As bacterial flagella are highly immunostimulatory, the lack of flagella in some pathogenic bacteria is generally believed to be an evolved mechanism for immune evasion. V. cholerae's extreme motility during infection[14] suggests that motility provides an evolutionary advantage to this pathogen that outweighs the downsides of stimulating the immune system. By linking increased motility to transmission, the phenotypes of the ΔmotV mutant highlight the potential evolutionary advantages of pathogen flagellation during enteric infection.

There is limited information regarding the role of V. cholerae motility and chemotaxis in the human intestine. In the feces of cholera patients it has been observed that V. cholerae are motile with reduced expression of several chemotaxis genes[18]. However, the microarray used in this study did not report information on motV or cheY-3. Likewise, there are limited studies of V. cholerae motility and chemotaxis in infant rabbits, which, like humans and infant mice, display Tcp-dependent V. cholerae colonization[17]. There have been several Tn-seq studies in infant rabbits, but the results disagree on the role of motV and cheY-3 during infection[27,36–38]. Fu, Waldor, and Mekalanos 2013 were the only study to validate a chemotaxis-related hit; they found that transposon-disruption of several chemotaxis-related genes, including motV, elevate V. cholerae infant rabbit colonization, and validated that a vspR mutant had greater than a 10-fold advantage in colonizing infant rabbits compared to WT[36]. However, other Tn-seq studies found that motV mutation has no impact[27,37] or is detrimental to infant rabbit colonization[38]. Further work is required to understand the role of V. cholerae motility and chemotaxis in the human intestine.

## Limitations of this study

The evolutionary benefits of motV are unclear. Presumably, V. cholerae motility and chemotaxis promote survival in the pathogen's natural environments. By contrast, we find that MotV's response to the stimuli encountered inside of the infant mouse SI is apparently maladaptive to survival/growth; the absence of motV promotes V. cholerae intestinal colonization and transmission between suckling mice, properties that seem beneficial for the propagation of the pathogen. These findings suggest a limitation of this otherwise valuable model. Possibly, the increased disease associated with the deletion of motV may increase mortality and be evolutionarily disadvantageous in humans, V. cholerae's natural host. However, regardless of the evolutionary logic for a gene like motV, our findings with the ΔmotV mutant illustrate the connection between intra-animal pathogenesis and inter-animal transmission.

The principal focus of this study was the characterization of the many infection-related phenotypes of a V. cholerae motV mutant, and how these phenotypes contribute to inter-animal transmission. Strains lacking cheY-3 activity have previously been extensively characterized[13–16,56–58]. The differences between cheY-3 and motV mutants are that only the deletion of motV increased virulence and inter-animal transmission. The molecular mechanisms accounting for the different infection phenotypes of these stains may result from differences in their respective intra-intestinal localization, virulence regulation, and/or colonization dynamics and represent an interesting area for future study. Regardless, the divergence of phenotypes between the motV and cheY-3 mutants provides a valuable tool for dissecting the mechanisms of V. cholerae transmission.

## Methods
### Biosafety statement
All work on V. cholerae was performed in Biosafety Level 2 (BSL2) facilities at the Brigham and Women's Hospital according to protocols reviewed and approved by the Brigham and Women's Hospital Institutional Biosafety Committee (protocol 2011B000082). All personnel

working with bacteria were trained with relevant safety and protocol-specific procedures.

## Risk-benefit analysis

This study was undertaken to enhance understanding of the cholera pathogen, which is an ongoing threat to global public health. This work included a transposon insertion site sequencing (Tn-seq) study in an infant mouse model to elucidate the pathogen factors that mediate enteric colonization. Such basic investigations are widely accepted to benefit public health by increasing our understanding of the mechanisms of pathogenesis.

Tn-seq screens study how disruption of bacterial genes by a transposon (knockout mutations) modify bacterial growth. Such loss-of-function (knockout) screens have been carried out in many different pathogens and models of infection and are generally accepted to be safe and low-risk because evolution has selected for the active form of genes. For *V. cholerae*, a pathogen that has co-evolved with its human host for millennia, loss-of-function (knockout) mutations generated within the lab are extremely unlikely to create a form of the bacteria that has not already been rejected by generations of evolutionary pressures. Loss-of-function (knockout) mutations are assumed safe because if they enhanced fitness of the pathogen in the natural host, then they would become more prevalent in wild populations. However, these mutations are not detectable in the wild population.

The vast majority of the Tn-Seq screens in animal models reported to date reveal genes on the right side of the volcano plot; deletion of these genes, like *motV*, increases colonization within the animal model. In general, the thinking in the field is that deletions that increase colonization in animal models are an artifact of adapting pathogens to non-natural hosts. Mice (and other experimental animals) are not natural hosts of *V. cholerae*. Thus, even though our findings suggest that deletion of *motV* causes increased colonization in the infant mouse model, this is not likely the case in humans. In general, although our work provides substantial new insight into the basic mechanisms by which motility contributes to pathogenicity, it does not provide a road map for the creation of a more pathogenic form of *V. cholerae*.

## Bacterial strains and growth conditions

Unless otherwise noted, bacteria were grown at 37 °C in lysogeny broth (LB) in liquid culture shaking at 200 rpm or on solid media containing 1.5% agar (weight/vol.). For the induction of cholera toxin, *V. cholerae* was grown in AKI broth (0.5% sodium chloride, 0.3% sodium bicarbonate, 0.4% yeast extract, and 1.5% Bacto peptone) for 4 h static, followed by 4 h with rotation at 37 °C[46,59]. All *V. cholerae* were derivatives of a spontaneous streptomycin (Sm) resistant mutant of a 2010 *V. cholerae* clinical isolate from Haiti[29]. Selection was accomplished by supplementing media with 200 µg/mL Sm. If required, antibiotics or other supplements were used in the following concentrations: carbenicillin (Cb), 100 µg/mL; kanamycin (Km), 50 µg/mL; diaminopimelic acid (DAP), 0.3 mM; sucrose (Suc), 0.2%; 5-bromo-4-chloro-3-indolyl-β-d-galactopyranoside (X-gal), 60 µg/mL. Bacterial stocks were stored at −80 °C in LB containing 25–35% (vol./vol.) glycerol. Strains used in this study are listed in Supplementary Data 2.

## Strain and plasmid construction

*V. cholerae* strains with deletions of *motV* and *cheY-3* were constructed using the vector pCVD442 and allelic exchange[60]. 700 bp homology arms upstream and downstream of the target open reading frame were amplified from the *V. cholerae* genome using the primers in Supplementary Data 3. The amplified region was then cloned into pCVD442 and introduced into *V. cholerae* by conjugation. Integration of the non-replicative plasmid was selected on Sm+Cb and double cross-overs removing the antibiotic-resistance cassette was selected for by growth in sucrose. PCR and amplicon sequencing confirmed deletion of the entire open reading (from start to stop codon).

The Δ*lacZ*[44] and Δ*ctxAB*[9] mutants were from previous studies. *V. cholerae lacZ::tdTomato* was previously described in ref. 14, and the same method and vector (pJZ111) was used to introduce *eGFP* into the *lacZ* loci of the Δ*motV* mutant to create Δ*motV lacZ::eGFP*. p*motV* was constructed by integrating the *motV* open reading frame into the pBR322 vector[61] downstream of a *ctxAB* promoter using the primers listed in Supplementary Data 3.

## Mice

Animal studies were conducted at the Brigham and Women's Hospital in compliance with the Guide for the Care and Use of Laboratory Animals and according to protocols reviewed and approved by the Brigham and Women's Hospital Institutional Animal Care and Use Committee (protocol 2016N000416).

CD1 (Crl:CD1(ICR)) dams with litters (mixed sex) were purchased from Charles River Laboratories (strain #022). Mice were housed in a biosafety level 2 (BSL2) facility under specific pathogen free conditions at 68–75 °F, with 30–50% humidity, and a 12 h light/dark cycle. CD1 infant mice at postnatal day 3-5 were intragastrically inoculated[42] with the indicated dose and strain of *V. cholerae* in 50 µL of LB. Except for survival and transmission experiments, the infant mice were housed separately from the dams in tissue-lined boxes during infection. For survival and transmission experiments, pups were returned to dams where they remained until morbidity or the predetermined end point of the experiment. Infant mice were euthanized by isoflurane inhalation followed by decapitation. Adult mice were euthanized at the end of the study by isoflurane inhalation followed by cervical dislocation.

## Infant mouse colonization

For colonization assays, infant mice were separated from their dams and intragastrically inoculated with the indicated dose and strain. Dose (colony forming units; CFU) was determined by retrospective serial dilution and plating on selective media containing Sm. At the indicated times (<24-hours post inoculation), pups were euthanized, and the SI was removed. Colonization burden was determined in either the whole SI, or the SI divided into 2 or 3 parts of equal length (proximal, medial, and distal). Tissue segments were further divided in half for experiments that combined measurements of bacterial localization (microscopy) and bacterial burden. To disassociate the bacteria from the tissue, organs were homogenized in LB using a bead beater (BioSpec Product, Inc) and 2 stainless-steel 3.2 mm beads. SI CFU was determined by serial dilution and plating on selective media containing Sm.

Competitive infections were performed the same as other colonization experiments, except that the inoculum was a ~1:1 mixture of the indicated strains. Blue/white CFU were determined by plating organ homogenates on LB + Sm/X-gal. Competetitive index was calculated by dividing the blue:white ratio from the SI by the blue:white ratio from the inoculum.

To measure diarrheaogenicity (diarrheal discharge, SI fluid accumulation, and weight loss), infant mice were individually housed for 18 h following inoculation. The mice were weighed immediately following inoculation and immediately before euthanasia to determine weight loss. Following infection, the weight of the pup was compared to the weight of the entire resected SI to determine SI fluid accumulation. The bedding was weighed at the start and end of the experiment to determine diarrheal discharge.

## Tn-seq

Experiments were performed with a WT Himar transposon library[38]. The library was expanded as a control on solid LB+Sm agar (LB library). For animal experiments, ~5 × 10⁷ CFU of the library was inoculated into infant mice. 18 h later, *V. cholerae* from the SIs of individual pups (Pup 1–3) or the pooled SIs of 10 mice from the same litter (Pooled litter) were outgrown overnight on LB+Sm agar. After outgrowth, cells were scraped off plates and frozen at -80 °C until processing.

Library preparation and sequencing were performed according to established protocols[30,62]. gDNA was harvested using the GeneJet gDNA Isolation Kit. gDNA was fragmented to 400 bp using an ultrasonicator. Fragment ends were repaired with New England Biolabs Quick Blunting Kit and A-tailing was performed with Taq DNA-polymerase. Adaptor sequences were ligated onto fragments with T4 DNA ligase and used to perform PCR to enrich fragments containing the transposon. 300–500 bp fragments were isolated using gel extraction. Libraries were then sequenced on an Illumina NextSeq.

Sequencing data was processed using CLC Genomics Workbench (version 12.0.2; Qiagen) and the R-based RTISAn pipeline, according to established protocols[27,30]. Trimming parameters (version 2.3): 3′ sequence – ACCACGAC; 5′ sequence – CAACCTGT; mismatch cost = 1; gap cost = 1; minimum internal score = 7; minimum end score = 4; discard reads <10 nt. Mapping parameters (version 1.6): reference – Haiti WT genome[63]; mismatch cost = 1; insertion cost = 3; deletion cost = 3; length fraction = 0.95; similarity fraction = 0.95; reads mapped to multiple locations were mapped randomly; global alignment. RTISAn was used to convert mapping files to a positional tally (TAtally) exclusively counting insertions at TA dinucleotides. To account for the stochastic loss of mutants caused by infection bottlenecks, the LB library TAtally (input) was resampled 100 times to approximate the diversity of the animal TAtally (output). The simulations of the input were compared to the output to derive an average fold-change and $P$ value (two-sided Mann–Whitney test).

To visualize the number of unique insertion sites per sample across a range of sampling depths, the TAtally was subjected to multinomial resampling beginning with the maximum read count of each sample and subsampling in two-fold lower increments. The number of unique insertions was calculated at each sampling depth. Prior to resampling, the TAtally was filtered by removing the bottom 1% of all reads to remove noise resulting from Illumina index sequence hopping.

## Microscopy

For microscopy, SI segments were retrieved as described for mouse colonization and then processed for cryosectioning[64]. Tissue samples (~2 cm) were fixed in PBS with 4% paraformaldehyde at 4 °C for 2–4 h, placed in PBS with 30% sucrose overnight at 4 °C, and then mounted in a medium containing a 1:2.5 ratio of 30% Sucrose: Optical Tissue Clearing medium (OCT). Mounted tissue pieces were snap-frozen in liquid nitrogen, stored at −20 °C for 1 h, then transferred to a −80 °C freezer until processing.

8–10 μm thick sections were cut on a Leica CM1860 cryostat and mounted on MAS adhesive slides (Matsunami Glass). Once samples were adhered to the slide, to preserve native fluorescence the samples were processed according to[64]. After sections were cut, tissue pieces were delineated with a PAP-pen, then slides were dried in the dark for 20 min at room-temperature. Ice-cold 4% PFA was then added for 8 min while slides were stored in a humid chamber. PFA was removed and slides were then washed with PBS and 3% BSA-PBS. Then, a 1:1000 dilution of Alexa Fluor 405-conjugated phalloidin (ThermoFisher) was added to each segment for 15 min, in a dark humid chamber. After 15 minutes, slides were washed with PBS and 3% BSA-PBS again and mounted with VectaShield Plus non-hardening antifade mounting medium (vector laboratories). Slides were stored for 1-hour flat at room-temperature, then sealed with clear nail polish and stored at 4 °C until imaging. Slides were imaged with a Nikon Ti2 Eclipse spinning disk confocal microscope using a 40x oil immersion lens with a numerical aperture of 1.40 and an Andor Zyla 4.2 Plus sCMOS monochrome camera. Image analysis was performed on the ImageJ (FIJI (2.14.0)) software using custom macros.

For quantification of the localization of Δ*motV*:WT *V. cholerae* ratios in competitive infection, the number of WT (red) and Δ*motV* (green) *V. cholerae* were counted in 50–60 100x100 micron images per location (proximal/medial/distal; crypt/top of villus), taken from 20-30 larger (244.16x244.16 micron) fields. When 0 WT *V. cholerae* were observed, the Δ*motV*:WT ratio was set to 35 and labeled as "ND" (not detected), as the highest ratio of Δ*motV*:WT *V. cholerae* with WT organisms detected was 32:1.

For quantification of crypt occupancy, the total number of crypts in each of 20 244.16 x 244.16 micron fields per time point, infection, and location was recorded, as was the number of crypts in each field containing ≥1 *V. cholerae*. Crypt occupancy was determined by dividing the number of occupied crypts by the total number of crypts per field.

## Soft agar motility

1 μl of an overnight culture of the indicated strains were injected into semisolid LB agar (0.3%; weight/vol.). Plates were cultured at 37 °C for 6-hours and the colony diameter was measured. A WT control was included on every plate.

## Mucus penetration

Mucus columns were prepared by adding 800 μL of 1% porcine mucus (Sigma) to a sterile syringe. *V. cholerae* were cultured overnight, diluted in saline (0.45% NaCl), and 50 μl were layered on top of the column. Columns were incubated for 30 minutes at 37 °C. To quantify penetration, 100 μl fractions were collected and plated to determine CFU.

## Bottleneck quantification

Barcoded *V. cholerae* libraries were created with the plasmid donor library pSM1[65]. The pSM1 donor library contains ~70,000 unique plasmids each with a random ~25 bp barcode carried within a Tn7 site-specific transposon. Barcodes were introduced into the *V. cholerae* Tn7-integration site by triparental mating of the recipient *V. cholerae* strain with the pSM1 donor library and a helper plasmid (pJMP1039) that expresses the Tn7 transposase. Transconjugants were then selected on LB Sm Km plates. After outgrowth, transconjugates were harvested from selective plates in LB glycerol (25-35%; weight/vol.) and stored in aliquots at -80 °C.

STAMP libraries were inoculated and SIs were retrieved as described for other infant mouse colonization assays. *V. cholerae* burden (CFU) was quantified by serial dilution and plating on selective media. *V. cholerae* from the remaining sample was outgrown on LB Sm and bacterial samples were harvested and stored at −80 °C in LB glycerol (25–35%; weight/vol.) until processing. Samples were processed for barcode sequencing and STAMPR analysis as described in Hullahalli et al. 2021[43]. Bacteria were boiled to release DNA and PCR was performed to amplify the barcode region and add sequencing adapters. Samples were sequenced on a NextSeq (Illumina). R and the STAMPR analysis pipeline[43] was used to demultiplex sequencing reads, trim, and map to the donor library pSM1. Founding population (Nr) was determined by STAMPR by comparing the number and frequency of barcodes recovered from the SI to a control sample outgrown from the animal inoculum. STAMPR scripts are available online at [https://github.com/hullahalli/stampr_rtisan].

## Infant mouse survival

Infant mouse challenge assays were performed as described in *Sit* et al., 2019[9]. Infant mice were orally inoculated with $10^6$ CFU of the indicated *V. cholerae* strain and returned to singly housed dams for maternal care. Survival was scored based on time from inoculation to morbidity. Body condition, diarrheal discharge, and temperature of infected mice were monitored every 4–6 h to determine the onset of symptoms. Symptomatic mice were monitored every 30-minutes until reaching morbidity, at which point the pups were immediately removed and euthanized. At the predetermined endpoint (30 h post inoculation), the remaining animals were scored as surviving and euthanized.

## Cholera toxin ELISA

*V. cholerae* strains were cultured in LB or AKI conditions. LB-culture: 8-hours in LB, with rotation, at 37 °C. AKI-culture[46,59]: 4 h static, followed by 4 h with rotation at 37 °C in AKI broth (0.5% sodium chloride, 0.3% sodium bicarbonate, 0.4% yeast extract, and 1.5% Bacto peptone).

GM1 ELISA was used to quantify the concentration of cholera toxin in cell-free supernatant samples according to established protocols[66]. Equal volumes of LB or AKI supernatants were serially diluted and known concentrations of purified cholera toxin were used as the standard. 96-well polystyrene microtiter plates were coated with GM1 ganglioside overnight, and 4 μg/ml fatty acid-free bovine serum albumin (BSA) was used to block the GM1-coated plates for 1 h at room temperature. Next, 260 μl of the supernatants were added to the wells in duplicate and incubated for 1 h at room temperature. Subsequently, a rabbit anti-CT polyclonal antibody (1:10,000) and then an HRP-linked goat ani-rabbit IgG antibody (1:5,000) were added to the wells and incubated for 1 h at room temperature each. For the development of the cholera toxin-antibody complex, tetramethylbenzidine (TMB) substrate solution (Thermo Fisher Scientific) was used according to the manufacturer's protocol. The color intensity in each well was measured at 485 nm in a plate reader. The absolute quantity of cholera toxin in the samples was estimated by comparison to the standard curve.

## Infant mouse transmission

Infant mice were randomly re-assorted to prevent litter bias. ~1/3 of each litter were inoculated with the indicated *V. cholerae* strain (seeds) and returned to foster-dams for maternal care with naïve littermates (contacts). Seeds and contacts remained with dams for 20-hours, at which point seeds and contacts were removed and euthanized. Transmission to contacts was determined by enumerating CFU in the SI. Contacts with 0 CFU in the SI were determined to be uninfected and contacts with ≥1 CFU were determined to be infected.

## Software and statistics

Data analysis was performed using CLC Genomics Workbench version 12.0.2 (Trim reads 2.3 and Read mapping 1.6 functions), R version 4.4.2 (RTISAn Tn-seq analysis[30]), GraphPad Prism, and Excel. Information regarding the number of samples and statistical tests are described in the figure legends. Geometric means, geometric standard deviations, and non-parametric tests were used for analyzing Tn-seq data, bacterial burden, bacterial competition, crypt occupancy, and founding population. Means, standard deviations, and parametric tests were used for comparisons of bacterial motility, diarrheal discharge, SI fluid accumulation, animal weight loss, and cholera toxin. Graphics and figures were prepared with BioRender, GraphPad Prism, and PowerPoint.

## Biological materials

Genetically modified strains of *Vibrio cholerae* were created for this study. Copies of these strains will be made available upon request from the corresponding author, Matthew K. Waldor.

## Reporting summary

Further information on research design is available in the Nature Portfolio Reporting Summary linked to this article.

## Data availability

Sequencing reads from Tn-seq are deposited in the Sequencing Read Archive (SRA) under accession no. PRJNA1247499. Source data are provided with this paper as a source data file. Source data are provided with this paper.

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

## Acknowledgements

We thank members of the Waldor lab for helpful discussions and feedback on the manuscript. We acknowledge our funding: Howard Hughes Medical Institute (M.W.). NIH grants P30 DK034854 (I.C.) and R01 AI042347 (M.W.). Fellowships T32 DK007477-37 (I.C.) and Zingl −2024HHMI (F.Z.). This article is subject to HHMI's Open Access to Publications policy. HHMI lab heads have previously granted a non-exclusive CC BY 4.0 license to the public and a sublicensable license to HHMI in their research articles. Pursuant to those licenses, the author-accepted manuscript of this article can be made freely available under a CC BY 4.0 license immediately upon publication. This manuscript is the result of funding in whole or in part by the National Institutes of Health (NIH). It is subject to the NIH Public Access Policy. Through acceptance of this federal funding, NIH has been given a right to make this manuscript publicly available in PubMed Central upon the Official Date of Publication, as defined by NIH.

## Author contributions

Conceptualization: I.C., R.D., M.W. Methodology: I.C., R.D., A.M., F.Z. Investigation: I.C., R.D., A.M., K.D., F.Z. Visualization: I.C., K.D. Writing the original draft: I.C., R.D., M.W. Reviewing and editing the manuscript: I.C., R.D., A.M., K.D., F.Z., M.W.

## Competing interests

The authors declare no competing interests.
