## [Transparent Peer Review file · Nature Communications]

***Vibrio cholerae* motility is associated with inter-animal transmission**

Corresponding Author: Professor Matthew Waldor

Version 0:

Reviewer comments:

Reviewer #1

(Remarks to the Author)

In this study by Campbell et al, a Tn-seq approach was used to understand the fitness of *V. cholerae* genes during infection in a mouse pup model of infection. This technique identified that transposon mutants in *motV* had increased fitness in vivo and this was validated by infecting mice with a defined mutant. *MotV* mutants induced increased transmission to naive hosts and this was attributed to increased diarrheal discharge from mice infected with the mutant. The results presented in this study are generally robust and well presented and the manuscript well written and easy to read.

Minor comments:

Lines 77-81 would benefit from some references.

L449 I think there is an error and this should be "Tukey" and not "Turkey".

Do individual mice consistently have higher insertions in *MotV* during the Tn-seq infection studies?

It would be helpful to state how statistical significance was obtained for figure 6A in the figure legend.

Did mice infected with the $\Delta\text{motV } \Delta\text{ctxAB}$ double mutant have reduced diarrheal discharge compared to the single *motV* mutant?

Reviewer #2

(Remarks to the Author)

This manuscript presents the identification and characterisation of *MotV* as a key determinant of *V. cholerae* colonisation in infant mice. Of significant interest is the ability of a ΔmotV mutant to circumvent the host colonisation bottleneck in infant mice and to increase virulence and transmissibility in this animal model. A very nicely, suitably and thoroughly designed and conducted study. The work is clearly described in this well presented manuscript. The data are convincing and support the interpretations and conclusions of the authors for *V. cholerae* colonisation in the infant mouse model.

However, there is limited reference to the authors' previous work and results with the *V. cholerae* Haiti Himar Tn library in the rabbit model, and limited comparison and contrast of the results obtained in each animal model. In particular, *MotV*, *CheY*, *CheA* nor *CheR* mutants were not enriched in the rabbit model in the previous work. Therefore the phenotypes observed in the study here and the conclusions of the authors are solely associated with the infant mouse model. This must be made more evident throughout the manuscript.

There are some further comments to improve the manuscript.

Comments:

Throughout the manuscript the restriction of the phenotypes observed and authors' conclusions to the infant mouse model should be made more evident.

In the introduction, the addition of text describing the results of the authors' previous work in the rabbit model with the *V. cholerae* Tn-seq library used in this study is necessary.

Similarly in the discussion, the comparison and contrast of the results presented here with the results of the authors' previous work in the rabbit model with the *V. cholerae* Haiti Himar Tn library used in this study is necessary. Perhaps a comprehensive comparison of the 2 studies could be included in the Results section.

To demonstrate reproducibility, in each figure legend state the number of independent experiments performed, and the number of replicates in each independent experiment.

Fig S2. Should the figure legend state swimming motility, rather than swarming motility?

I. 468-476. How were the homolog arms and motV ORF produced? State the bp deleted from the motV gene

I. 477. change to "open reading frame"

Addition information is needed on construction of each of the strains listed in Table S1 as generated In This Study.

Reviewer #3

(Remarks to the Author)

Vibrio cholerae is the causative agent of highly infectious disease, cholera. Authors first carried out Tnseq in infant mouse infection model, and the results suggested that *V. cholerae* motV mutant has better fitness during the infection. Previously, Δ motV mutant is known to be defective in tumbling. While it shows motility defect in conventional semisolid agar-based assay, cells in fact can 'swim' in liquid environment although only straightly. Relationship between chemotaxis/motility and colonization/virulence has been studied in many different pathogens, sometimes led controversial concepts.

Here, authors carried out panel of infection experiments for further characterization of the *V. cholerae* Δ motV mutant. Results strongly indicated that Δ motV mutant exhibited elevated colonization in small intestine (SI), particularly in the proximal segment and at crypts in early time points. Δ motV mutant's better colonization was also explained by its looser bottleneck analyzed by NGS-based STAMPR. Furthermore, Δ motV mutant showed increased virulence and importantly, transmissibility.

Overall, experiments are well designed and executed (there are no unnecessary animal experiments.) Results were appropriately presented and analyzed (including statistics), and conclusions are fairly drawn from the results. Particularly, bottleneck and inter-animal transmission experiments are remarkable and they provided key results (I'm surprised that transmission experiment was not established before).

I have two concerns (shown below) but they are indeed pointed out by the authors in the 'limitation of this study'. Otherwise, this article will be very interesting and the knowledge is useful for our further understanding on the infection cycle of not only *V. cholerae* but also other enteric pathogens.

--

(A) Results clearly demonstrated that Δ motV mutant of *V. cholerae* is much potent to cause disease. It is therefore puzzling that 'wild type' strains, including the one used in this study which was originally isolated from cholera outbreak in Haiti, are motV+.

Are authors aware any clinical strains that are naturally mutated in motV?

It is very possible that Δ motV mutant has worse fitness in environmental reservoirs.

I think it will be intriguing to extend discussion regarding to motility in mucus. From the results, it seems that although viscous, mucus provides environment rather similar to liquid than semisolid agar. Furthermore, there is a report (by the author's group) showing that mucin, the main component of mucus layer, is a key inhibitor of *V. cholerae* colonization particularly in the proximal SI (Ref 15/29, see below) where Δ motV can colonize much better than the wild type. It is then tempting to speculate that Δ motV mutant shows better motility in mucus. Are there any established in vitro settings that recapitulate the mucus environment?

(B) Δ cheY and Δ motV mutants both show defect in tumbling. They also share same phenotype of better colonization in SI, in terms of CFU and competition. However, authors showed marked difference between Δ motV and Δ cheY mutants in the downstream reactions: diarrhea, mortality, and transmissibility.

It is quite interesting to know the molecular basis that made the difference.

The limiting thing is that, in this study authors showed detailed localization/colonization pattern of Δ motV mutant over the time of infection, as well as bottleneck. Although extensive studies are carried out to characterize Δ cheY mutant, they are relatively old so that the counterparts of Δ cheY mutant are missing.

I must add that it is not fair at all to ask additional experiments of Δ cheY mutant for this article. However, if there are any data available that would enrich the discussion.

I may propose to include in the discussion:

(1) Δ cheY in other bacteria are often defective in infection (Ref 14).

(2) MotV is restricted in *Vibrio* and closely related bacteria, while chemotaxis proteins (including CheY) are very well

conserved (this information can be also useful to discuss evolutionary benefits of motV).

Miscellaneous points

(c) *V. cholerae* encodes multiple chemotaxis genes such as 5 cheYs (Ref 46). It may be better to precise the gene (e.g. cheY1) or define when mentioned first.

(d) There are a few doubled references: 15 = 29, 16 = 17, 22 = 54.

Version 1:

Reviewer comments:

Reviewer #1

(Remarks to the Author)

The authors have made changes to the manuscript that have addressed the questions raised previously. The added figures and text provide clarity and add robustness to their original findings and conclusions.

Reviewer #2

(Remarks to the Author)

The authors have appropriately and thoroughly addressed the comments of the reviewers in their revised manuscript.

Reviewer #3

(Remarks to the Author)

I believe authors carefully addressed my comments and those of other reviewers. I particularly appreciate authors carried out mucus penetration assay and extended the discussion.

Possibly a typo in Figure 3 legend (p9) : *D*, n = 20 fields... » *B*, n = 20 fields...

Response to Referees

As outlined below in our point-by-point responses, we have addressed all the referees' comments with new experiments, analyses, and modifications to the text. These modifications include testing the motility of the *motV* mutant in mucus, adding 3 new figure panels, and a more extensive discussion regarding the roles of *V. cholerae* chemotaxis and motility in humans and the infant rabbit model of cholera. The referees' thoughtful questions and suggestions have improved the manuscript, and we thank you for your time and efforts in considering our work. We look forward to the assessment of our revised manuscript.

Point-by-point response (review in black, response in blue)

Reviewer #1 (Remarks to the Author):

In this study by Campbell et al, a Tn-seq approach was used to understand the fitness of *V. cholerae* genes during infection in a mouse pup model of infection. This technique identified that transposon mutants in *motV* had increased fitness in vivo and this was validated by infecting mice with a defined mutant. *MotV* mutants induced increased transmission to naive hosts and this was attributed to increased diarrheal discharge from mice infected with the mutant. The results presented in this study are generally robust and well presented and the manuscript well written and easy to read.

We thank the reviewer for their positive comments.

Minor comments:

Lines 77-81 would benefit from some references.

Thank you for identifying this omission. We added references to recent reviews at the first mention of transposon insertion screens. Furthermore, we cite the 4 studies from 2009 that first used Tn-seq to identify bacterial fitness determinants within the host and host-related environments.

L449 I think there is an error and this should be "Tukey" and not "Turkey".

Thank you for identifying this error. We have corrected all incorrect references from "Turkey" to "Tukey."

Do individual mice consistently have higher insertions in *MotV* during the Tn-seq infection studies?

Yes. When we analyze the data from 3 individual pups separately, the data is noisier than the pooled litter, but there is a clear enrichment in insertions in *motV* in all 3 animals. We have included this new analysis as a new panel in the supplement (Fig S1C).

It would be helpful to state how statistical significance was obtained for figure 6A in the figure legend.

Thank you for identifying this omission. The statistical test is Fischer's exact test. We have added a description of the test to the legend of Fig 6A.

Did mice infected with the Δ motV Δ ctxAB double mutant have reduced diarrheal discharge compared to the single motV mutant?

Yes. In addition to decreasing SI fluid accumulation and animal weight loss, the deletion of *ctxAB* reduced the diarrheal discharge of the Δ motV mutant. The *V. cholerae* inoculum is dyed green to assist with intragastric gavage, and subsequent dye discharge stains the bedding, visualizing diarrheal discharge. There was no visible discharge in the Δ motV Δ ctxAB double mutant. We have added this data to Fig S6.

Reviewer #2 (Remarks to the Author):

This manuscript presents the identification and characterisation of MotV as a key determinant of *V. cholerae* colonisation in infant mice. Of significant interest is the ability of a Δ motV mutant to circumvent the host colonisation bottleneck in infant mice and to increase virulence and transmissibility in this animal model. A very nicely, suitably and thoroughly designed and conducted study. The work is clearly described in this well presented manuscript. The data are convincing and support the interpretations and conclusions of the authors for *V. cholerae* colonisation in the infant mouse model.

We thank the reviewer for the positive assessment.

However, there is limited reference to the authors' previous work and results with the *V. cholerae* Haiti Himar Tn library in the rabbit model, and limited comparison and contrast of the results obtained in each animal model. In particular, MotV, CheY, CheA nor CheR mutants were not enriched in the rabbit model in the previous work. Therefore the phenotypes observed in the study here and the conclusions of the authors are solely associated with the infant mouse model. This must be made more evident throughout the manuscript.

Thank you for identifying this area for clarification. We have added additional text comparing the findings reported here with those from previous studies in infant rabbits. These additions will help the reader contextualize the results from the current study.

As we discuss in the revised manuscript, comparisons between our study and the 4 previously published Tn-seq screens in infant rabbits are somewhat difficult because the results of the infant rabbit screens differ. For example, our study recovered an increased number of insertions in *motV* from infant mice. Similarly, an increased number of insertions in *motV* (VC1909) were recovered in a Tn-seq study of a Peruvian El Tor O1 clinical isolate (C6706) in infant rabbits (Fu, Waldor, and Mekalanos 2013). However, 2 infant rabbit Tn-seq studies (Kamp, et al., 2013; Pritchard, et al., 2014) using strains E7946 and C6706, respectively, reported that *motV* mutation is neutral. Furthermore, a subsequent Tn-seq study using the same library as our manuscript found that fewer mutants in *motV* were recovered from infant rabbits (Hubbard et al., 2018). Unfortunately, none of these studies validated or explored the *motV* result.

Examining exclusively Hubbard et al., 2018 (which used the same *V. cholerae* as our study), it is notable that there appears to be opposite results for chemotaxis and motility as observed in mice. In our study, mutations in *motV*, *cheA*, *cheR*, and *cheY* all elevate *V. cholerae* colonization of infant mice, while these mutations cause a minor colonization defect in the infant rabbit study. Likewise, in our study genes involved in flagellar assembly (e.g., *fli* operon, *flg* operon, *motB*, *motXY*) are infant mouse colonization factors, while mutations in the same genes cause a minor or no colonization defect in the 2018 infant rabbit study. Our findings in infant mice mirror many previous studies on the roles of

chemotaxis and flagellation in *V. cholerae* colonization of infant mice. Thus, flagellation and chemotaxis may play different roles in colonization in these 2 models.

Both infant mice and infant rabbits are valuable tools for understanding human *V. cholerae* pathogenesis. Both models agree with our limited information on the *V. cholerae* genes that modify the colonization of humans. However, in the absence of human study data for most *V. cholerae* genes, it is difficult to resolve which model is more relevant to the human context.

Ultimately, understanding the biological bases of the discrepancies observed between infant mice and rabbits could further disentangle the complex relationship between motility, chemotaxis, and *V. cholerae* colonization. However, this work goes beyond the scope of the current study, which demonstrates that Tn-seq is feasible in infant mice, a more accessible model for the cholera research field. Moreover, our study provides the first in-depth analysis of the relationship between motility and cholera transmission between infant mice.

There are some further comments to improve the manuscript.

Comments:

Throughout the manuscript the restriction of the phenotypes observed and authors' conclusions to the infant mouse model should be made more evident.

Thank you for identifying this area for clarification. The model used in this study (infant mice) is identified in the abstract's second sentence, the introduction's second paragraph, and every experiment and result. In addition, we have made the following additions to the text:

(1) We have added information to the introduction about what is known regarding the roles of *V. cholerae* chemotaxis and motility in humans and infant rabbits.

(2) We clarified in the results section that the importance of flagellation has been previously tested only in infant mice.

(3) We have added a statement to the initial description of *motV* in the results section indicating that the role of *motV* has not been tested in human or infant rabbits. Furthermore, we indicate that there are conflicting results regarding *motV* from previous infant rabbit Tn-seq experiments.

(4) We added a paragraph discussing what is known regarding *V. cholerae* motility and chemotaxis in humans and infant rabbits to the discussion.

In the introduction, the addition of text describing the results of the authors' previous work

in the rabbit model with the *V. cholerae* Tn-seq library used in this study is necessary.

We have added text to the introduction clarifying what is known from previous work regarding the roles of *V. cholerae* motility and chemotaxis in humans and infant rabbits. We have kept the description of previous infant rabbit studies in the introduction brief; we believe that the disagreement between previous studies would confuse the reader and have little value without further experimentation, especially in the introduction before we discuss the findings of this study. We also discuss previous infant rabbit studies and what is known in humans about the genes from this study in more detail in the results and discussion sections.

Similarly in the discussion, the comparison and contrast of the results presented here with the results of the authors' previous work in the rabbit model with the *V. cholerae* Haiti Himar Tn library used in this study is necessary. Perhaps a comprehensive comparison of the 2 studies could be included in the Results section.

We have added a discussion paragraph on the limited information that previous studies have provided regarding *V. cholerae* motility and chemotaxis in infant rabbits. We conclude that little is known about the role of *V. cholerae* motility and chemotaxis in infant rabbits due to a lack of agreement between studies and a lack of validation.

A comprehensive comparison of the genetic determinants of *V. cholerae* colonization in infant mice and infant rabbits may uncover interesting biology; in our view, such an analysis is beyond the scope of our current study.

To demonstrate reproducibility, in each figure legend state the number of independent experiments performed, and the number of replicates in each independent experiment.

Thank you for identifying this area for clarification. We have added further descriptions to the legends.

Fig S2. Should the figure legend state swimming motility, rather than swarming motility?

We have changed the legend to "Motility of indicated *V. cholerae* strains in soft agar plates."

l. 468-476. How were the homolog arms and *motV* ORF produced? State the bp deleted from the *motV* gene

We added further information regarding the strains constructed for this study, including the primer sequences. The *motV* deletion spanned the entire open reading frame (from start to stop codon).

l. 477. change to "open reading frame"

Done, thank you.

Addition information is needed on construction of each of the strains listed in Table S1 as generated In This Study.

We added further information regarding the strains constructed for this study.

Reviewer #3 (Remarks to the Author):

Vibrio cholerae is the causative agent of highly infectious disease, cholera. Authors first carried out Tnseq in infant mouse infection model, and the results suggested that *V. cholerae* *motV* mutant has better fitness during the infection.

Previously, Δ *motV* mutant is known to be defective in tumbling. While it shows motility defect in conventional semisolid agar-based assay, cells in fact can 'swim' in liquid environment although only straightly. Relationship between chemotaxis/motility and colonization/virulence has been studied in many different pathogens, sometimes led controversial concepts.

Here, authors carried out panel of infection experiments for further characterization of the *V. cholerae* Δ *motV* mutant. Results strongly indicated that Δ *motV* mutant exhibited elevated colonization in small intestine (SI), particularly in the proximal segment and at crypts in early time points. Δ *motV* mutant's better colonization was also explained by its looser bottleneck analyzed by NGS-based STAMPR. Furthermore, Δ *motV* mutant showed increased virulence and importantly, transmissibility.

Overall, experiments are well designed and executed (there are no unnecessary animal experiments.) Results were appropriately presented and analyzed (including statistics), and conclusions are fairly drawn from the results. Particularly, bottleneck and inter-animal transmission experiments are remarkable and they provided key results (I'm surprised that transmission experiment was not established before).

We thank the reviewer for the positive assessment of our work.

I have two concerns (shown below) but they are indeed pointed out by the authors in the 'limitation of this study'. Otherwise, this article will be very interesting and the knowledge is useful for our further understanding on the infection cycle of not only *V. cholerae* but also other enteric pathogens.

--

(A) Results clearly demonstrated that Δ *motV* mutant of *V. cholerae* is much potent to cause disease. It is therefore puzzling that 'wild type' strains, including the one used in this study which was originally isolated from cholera outbreak in Haiti, are *motV*⁺.

Are authors aware any clinical strains that are naturally mutated in *motV*?

It is very possible that Δ *motV* mutant has worse fitness in environmental reservoirs.

Thank you for discussing this interesting and important point. We are not aware of any clinical isolates lacking *motV*, suggesting that this gene has a role somewhere in the pathogen's natural lifecycle. However, it is currently unclear in which environments *V. cholerae* benefits from *motV*. *MotV* may have a role in *V. cholerae*'s non-host reservoir and/or in the human intestine. Unfortunately, there is limited information regarding the role

of *V. cholerae* motility and chemotaxis in the human intestine, making it difficult to determine if the infant mouse $\Delta motV$ phenotypes would be observed during human infection. We have added additional statements to the introduction and discussion about the lack of information regarding the roles of *V. cholerae* chemotaxis and motility in humans to alert readers to this limitation of our study.

I think it will be intriguing to extend discussion regarding to motility in mucus. From the results, it seems that although viscous, mucus provides environment rather similar to liquid than semisolid agar. Furthermore, there is a report (by the author's group) showing that mucin, the main component of mucus layer, is a key inhibitor of *V. cholerae* colonization particularly in the proximal SI (Ref 15/29, see below) where $\Delta motV$ can colonize much better than the wild type. It is then tempting to speculate that $\Delta motV$ mutant shows better motility in mucus. Are there any established in vitro settings that recapitulate the mucus environment?

Thank you for raising this interesting point. It was previously demonstrated that the *V. cholerae* flagellum is required for the bacteria to penetrate into the mucus layer (Liu et al., 2008, DOI: 10.1073/pnas.0802241105). The increased penetration of the $\Delta motV$ mutant into the crypts could indicate that this mutant can swim better in mucus and/or is better at reaching the mucus layer. We used an established mucus penetration assay and found that deletion of *motV* modestly increases motility in mucus *in vitro*. We have added these data as Fig S2D.

It is conspicuous that mucus disruption and *motV* deletion both increase *V. cholerae* colonization of the proximal SI. This correlation, along with the increased motility of the $\Delta motV$ mutant in mucus, may indicate that the relaxed bottleneck experienced by the $\Delta motV$ mutant reflects increased penetration into the mucus in the proximal SI. We have added additional text to the discussion to explore this hypothesis.

(B) $\Delta cheY$ and $\Delta motV$ mutants both show defect in tumbling. They also share same phenotype of better colonization in SI, in terms of CFU and competition. However, authors showed marked difference between $\Delta motV$ and $\Delta cheY$ mutants in the downstream reactions: diarrhea, mortality, and transmissibility.

It is quite interesting to know the molecular basis that made the difference.

The limiting thing is that, in this study authors showed detailed localization/colonization pattern of $\Delta motV$ mutant over the time of infection, as well as bottleneck. Although extensive studies are carried out to characterize $\Delta cheY$ mutant, they are relatively old so that the counterparts of $\Delta cheY$ mutant are missing.

I must add that it is not fair at all to ask additional experiments of $\Delta cheY$ mutant for this article. However, if there are any data available that would enrich the discussion.

Unfortunately, we could not differentiate the molecular mechanism causing the differences in $\Delta cheY$ and $\Delta motV$ infection phenotypes. The mutants display different

phenotypes in terms of diarrheagenicity and inter-animal transmission. However, according to our study they have similar phenotypes regarding colonization and cholera-toxin production (in culture). Similarly, previous studies indicate that the deletion of *cheY* causes similar phenotypes as we find for the deletion of *motV* in terms of swimming behavior, infectivity, and localization along the distal-proximal SI axis. Perhaps more direct comparisons between these strains could find further differences indicating mechanism, but our experiments did not reveal these results. We discuss the similarities and differences between the $\Delta cheY$ and $\Delta motV$ mutant phenotypes in the sections “Discussion” and “Limitations of this study”.

I may propose to include in the discussion:

(1) $\Delta cheY$ in other bacteria are often defective in infection (Ref 14).

Added to the discussion, thank you.

(2) MotV is restricted in *Vibrio* and closely related bacteria, while chemotaxis proteins (including CheY) are very well conserved (this information can be also useful to discuss evolutionary benefits of *motV*).

Added to the first description of *motV*, thank you.

Miscellaneous points

(c) *V. cholerae* encodes multiple chemotaxis genes such as 5 *cheYs* (Ref 46). It may be better to precise the gene (e.g. *cheY1*) or define when mentioned first.

Thank you. We have added precise identifications to the text.

(d) There are a few doubled references: 15 = 29, 16 = 17, 22 = 54.

Thank you. The duplicated references have been consolidated.